# "The Ability to Go Out into the World Is the Most Important Thing"—A Qualitative Study of Important Exercise Outcomes for People with Lung Cancer

Amy Bowman [1,2], Linda Denehy [1,3], Cassie E. McDonald [1,4] and Lara Edbrooke [1,3,*]

1   Department of Physiotherapy, The University of Melbourne, Melbourne, VIC 3053, Australia; amy.bowman@student.unimelb.edu.au (A.B.); l.denehy@unimelb.edu.au (L.D.); ca.mcdonald@alfred.org.au (C.E.M.)
2   Department of Physiotherapy, Peter Mac Callum Cancer Centre, Melbourne, VIC 3000, Australia
3   Department of Health Services Research, Peter Mac Callum Cancer Centre, Melbourne, VIC 3000, Australia
4   Allied Health, Alfred Health, Melbourne, VIC 3004, Australia
*   Correspondence: larae@unimelb.edu.au; Tel.: +61-03-9035-4215

**Abstract:** Whilst existing quantitative research identifies outcomes believed to be important by researchers and clinicians, it may neglect outcomes that are meaningful to patients. This study aimed to explore the outcomes of exercise that are important to people with lung cancer and their carers. Data collection involved a qualitative methodology including semi-structured interviews and focus groups. Question guide development was informed by the International Classification of Functioning (ICF) framework. Data were analyzed by two researchers with NVivo (v12) software using a conventional content analysis process, followed by directed content analysis to map outcomes to the ICF. Conduct and reporting adhered to COREQ guidelines. Fifteen participants provided data. Most participants had received their diagnoses 24 months prior to study involvement ($n = 9$), and one-third had completed treatment ($n = 5$). Important outcomes were reported by participants across all domains of the ICF: activity and participation ($n = 24$), body function ($n = 19$), body structure ($n = 5$), environmental factors ($n = 5$), and personal factors ($n = 1$). Additional code categories pertained to the impacts of non-cancer factors such as age, frailty, and comorbidities; identifying barriers to exercise; and individualizing outcome measures. Clinicians and researchers should consider selecting outcomes from all relevant domains of the ICF, with a focus on the activity and participation domain, in addition to non-cancer factors such as ageing, frailty, and co-morbidities. Feedback should be provided to patients following outcome measures collection and reassessment.

**Keywords:** lung cancer; exercise; outcome measures; qualitative; international classification of functioning





## 1. Introduction

Lung cancer is the leading cause of cancer mortality and morbidity in Australia, accounting for 18% of cancer-related deaths and 18.9% of the cancer-related burden of disease [1]. Associated with a higher symptom burden than other tumor types [2], people with lung cancer have high rates of functional decline and reduced health-related quality of life throughout the disease's trajectory [3]. Exercise training improves functional fitness, reduces treatment side effects, decreases symptom burden, and improves health-related quality of life across the lung cancer care continuum [4]. Due to these health benefits, exercise guidelines recommend exercise for people with cancer across the care continuum [5,6].

Oncology exercise guidelines recommend comprehensive assessment before, during, and following exercise programs to tailor interventions and monitor patient progress [7]. Clinical outcome measures are defined as any measurement or tool that can be used to assess a patient's current level of health, function, or movement [8]. They include a range of approaches, such as patient-, observer-, or clinician-reported outcomes, in addition

to objectively measured performance outcomes, and they are used to both directly and indirectly measure health aspects that are meaningful to patients using categorical or continuous ratings [8,9]. For outcome measures to be meaningful in directing patient care, measuring research impact or informing policy, they must reflect patient, clinician, and health service priorities [10].

The International Classification of Functioning (ICF) framework provides a standard language and conceptual basis for understanding and describing health and disability outcomes across a comprehensive set of domains [11]. This framework is founded upon the understanding that an individual's health and disability status are influenced not only by their medical condition but also by a confluence of physical, social, and environmental factors. The ICF comprises three primary domains related to functioning and disability and two contextual domains, namely environmental and personal factors, which play a significant role in determining an individual's health and disability status [11]. Within each ICF domain, there is a hierarchy of levels to provide further detail when classifying aspects of an individual's functioning or health condition. Each level expands on specific details, for example, within the activity and participation domain, there is a level one chapter for mobility, a second level for walking and moving, a third level for walking, and a fourth level for walking short distances. The ICF framework is used globally for outcome evaluation in the clinical rehabilitation setting [12], and it is used in research to classify and assess outcome measures used in patient assessment [13,14]. In lung cancer, no studies have applied the ICF to comprehensively examine which outcomes are important to patients and their caregivers.

There has been a growing focus on using outcomes perceived to be important to patients, also known as patient-centered outcomes, which incorporate patients' perspectives and experiences regarding the impact of healthcare interventions. Whilst existing quantitative research identifies outcome measures believed to be important by researchers and clinicians, it may neglect outcome measures that are meaningful to patients [15]. Systematic reviews in non-oncology populations report that patient-centered outcomes are under-represented in clinical trials [16,17]. The development of a patient-centered core outcome set is currently being conducted in the fields of exercise and lung cancer in Australia [18]. Work is underway by the European Health Outcomes Observatory initiative to implement a patient-centered core outcome set for lung cancer to measure the impacts of novel treatments and cancer burden in Europe [19]. This suggests that globally, there is increasing interest in understanding which outcomes are important to patients and applying these in practice. A limitation of the work to date is that it has only involved patients as advisory group members providing specific feedback on a set of outcomes, yet it has not included an exploration of patient perspectives during the development phase of the preliminary set of outcomes (i.e., through qualitative focus groups or interviews with patient participants).

A rich and detailed understanding of patient perspectives on important exercise outcomes should inform outcome selection in clinical practice and research to optimize the delivery of patient-centered care. Combining findings from quantitative consensus-based research and qualitative interviews and focus groups enhances understanding of patient-relevant outcome measures. Currently, there is no evidence reporting on what outcomes before, during, and after exercise are important to people with lung cancer or caregivers of people with lung cancer. Therefore, the primary aim of this study was to explore the outcomes of exercise programs that are important to people with lung cancer and caregivers of people with lung cancer. The secondary aim was to assess the frequency of outcomes related to domains within the ICF framework by participants.

## 2. Materials and Methods

### 2.1. Ethical Considerations

This study was approved by the University of Melbourne Human Research Ethics Committee (Project Number: 2021-22492-20870-3). Participants were provided with a consent form and plain language statement prior to attending a focus group or interview.

In the week prior to participation, a researcher called participants to reiterate and clarify consent processes and answer any questions prior to attending a focus group. Informed verbal consent was obtained from all participants to allow recording to commence, and further informed verbal consent was obtained for study participation from all participants once recording began. Participants were informed that their name would be displayed to other participants during the focus group and were given the opportunity to use an alias.

## 2.2. Research Team and Reflexivity

The all-female research team consisted of four physiotherapists. The primary researcher (A.B.) was an oncology physiotherapist guided by the expertise of PhD-qualified research professionals with experience in lung cancer, exercise, and qualitative research (L.E. and L.D.) and an experienced PhD-qualified qualitative researcher (C.M.) with experience outside of cancer and exercise research. All researchers acknowledged the influence of their own past experiences and perspectives at each stage of data collection, analysis, and result synthesis. Researchers openly discussed and documented reflections on how their past experiences and assumptions may have influenced their interpretation of the data following each interview and focus group. Two researchers (A.B. and L.E.) were involved at each data analysis stage and met regularly to discuss data interpretation. They considered how their preconceived ideas and clinical experience working in oncology may have influenced the construction of meaning from the data.

Focus group and interview facilitators were not known to the participants before involvement in this study. Participants were provided with a brief background of the facilitators' clinical and research experience at the commencement of the focus group or interview.

## 2.3. Study Design

A qualitative methodology aligning with a pragmatic epistemological approach [20] was employed to engage consumers in a flexible and iterative research process with practical implications. Semi-structured focus groups and interviews using teleconferencing modes were conducted to provide a rich and detailed understanding of patients' experiences and perspectives [21]. Offering interviews scheduled to an individual person's availability and using virtual modes to conduct interviews or focus groups increases the participation of people who may not be able to travel or have limited time [21].

Participants were invited to attend a single virtual focus group using the Zoom Video Communications, Inc.© (v5.0.1) teleconferencing platform [22]. Individual interviews were conducted for participants who were unable or declined to attend a focus group (videoconferencing) or were unable to use the technology (telephone call).

Question guide development was informed by the ICF framework [11]. The question guide was piloted twice with a consumer with lived experience and a healthcare professional working in the area of exercise oncology for clarity, with modifications made based on feedback and reflection. The question guide is provided in Supplementary Materials Section S1.

## 2.4. Participants, Recruitment, and Samples

Participants were people with lung cancer or the carers of people with lung cancer. This study's details were advertised in relevant consumer e-mail newsletters, lung cancer consumer Facebook groups, and shared on researchers' twitter feeds. Clinicians across the Victorian Integrated Cancer Services network were provided with the study's details to share with potential participants. Participant sampling was consecutive. Researchers reflected on and discussed data quality during data collection and preliminary data analysis and determined the sample size when the collected data were rich and represented a broad range of views on exercise outcome measures [23]. Pragmatic considerations related to the number of individuals who gave consent to participate were considered when recruiting the sample.

*2.5. Data Collection*

Data collection involved semi-structured focus groups created via videoconferencing (A.B., C.M., L.E., L.D.). Individual interviews were conducted via videoconferencing or telephone for participants who preferred this option or could not attend focus groups (A.B., C.M.). Focus groups and individual interviews were audio- and video-recorded, with recordings stored electronically for analysis. Researchers made field notes on observations and reflections made during data collection and analysis to record and deepen engagement with data.

*2.6. Data Analysis*

Descriptive statistics were used to describe the demographic and clinical characteristics of the sample, using n (%) for categorical variables and either mean (SD) or median (IQR) for continuous variables, depending on their distribution. Focus group and interview recordings were de-identified, transcribed verbatim (A.B.), and independently cross-checked by a second researcher (L.E.).

Data were analyzed using QSR International NVivo 12© software [24]. Firstly, data were analyzed using a conventional inductive content analysis approach [25]. Conventional content analysis was completed to enable a rich engagement with the breadth of outcomes represented by the participants. This analysis involved six-phases: (1) familiarization with the data set; (2) the initial coding of words, phrases, or segments that are relevant to the research question; (3) refining codes and organizing into clusters; (4) developing and refining categories; (5) the synthesis and interpretation of the data; and (6) reporting the results. Prior to commencing phase five, a directed content analysis was performed [25]. Relevant words, phrases, or segments of the text were deductively mapped to the ICF framework [11]. The ICF framework was used for directed content analysis to represent outcome measures using a universally understood and clinically relevant framework. Following this, a summative content analysis approach was conducted in the form of a word frequency count and a code frequency count. Words describing or referring to ICF domains, level one chapters, and second-level descriptions were counted, and final counts were compared to the number of words relating to the ICF domains, level one chapters, and second-level descriptions (see Supplementary Materials Figure S1 [25]). A count was conducted of each word attributed to a certain code, including codes that were attributed to more than one ICF domain. Data analysis commenced during the data collection period. Summative content analysis was completed to compare the frequency of codes with the frequency of words coded to the importance of specific outcomes. This step was performed to ensure there were no large disparities between the volume of discussion and the number of outcome codes related to domains within the ICF.

This study employed both inductive and deductive content analysis approaches to facilitate a pragmatic and targeted exploration of consumers' views and beliefs within a predefined framework while maintaining openness to emergent themes and unexpected findings.

*2.7. Rigour*

Synthesized member checking was undertaken to provide participants with the opportunity to engage with, reflect on, and add to the experiences, opinions, and ideas as interpreted by researchers (A.B. and L.E.) [26]. This study was conducted and reported according to the consolidated criteria for reporting qualitative research (COREQ) checklist [27].

### 3. Results

*3.1. Demographics*

The flow of participants through this study is shown in Figure 1. Seventeen potential participants contacted the research team to express interest, two decided not to participate in a focus group due to time (*n* = 1) and caring responsibilities (*n* = 1), and fifteen (88%) participated in one of four focus groups (*n* = 10) or individual interviews (*n* = 5). Participants

were people with lung cancer (*n* = 14) and a carer of a person with lung cancer (*n* = 1), female (*n* = 12), >2 years post-diagnosis (*n* = 9), and receiving active treatment for lung cancer (*n* = 10). Further demographic and clinical characteristics are reported in Table 1.

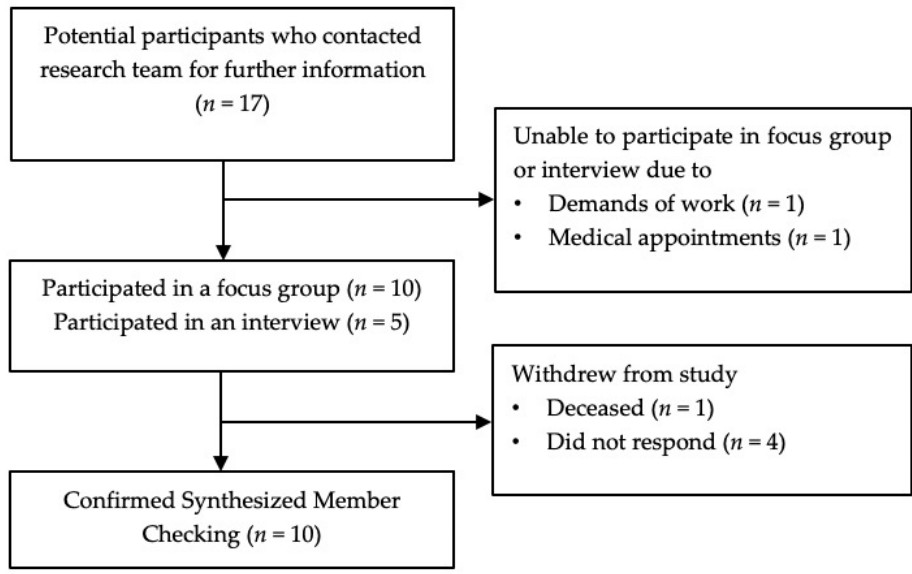

**Figure 1.** Flow through the study.

**Table 1.** Participants' demographic and clinical characteristics.

| Variable | N (%) |
|---|---|
| Age (years), mean (SD) | 55.6 (10.6) |
| Sex (male) | 3 (20) |
| Diagnosis of lung cancer | 14 (93.3) |
| Time since diagnosis of lung cancer | |
| 1–5 months | 1 (7.1) |
| 6–11 months | 2 (14.3) |
| 12–23 months | 2 (14.3) |
| ≥24 months | 9 (64.3) |
| Cancer treatment(s) received by participants with lung cancer | |
| Chemotherapy, radiotherapy, and immunotherapy | 5 (35.7) |
| Chemotherapy, radiotherapy, surgery, and immunotherapy | 4 (28.6) |
| Surgery, chemotherapy, and radiotherapy | 1 (7.1) |
| Surgery | 2 (14.3) |
| Chemotherapy | 1 (7.1) |
| Targeted therapy (EGFR inhibitor) | 1 (7.1) |
| Time since last cancer treatment received | |
| Currently receiving | 9 (64.3) |
| 1–5 months | 2 (14.3) |
| 6–11 months | 1 (7.1) |
| 12–23 months | 2 (21.4) |
| Rural residential status | 8 (53.3) |
| State of residence | |
| Victoria | 11 (73.3) |
| Queensland | 4 (26.7) |

**Table 1.** *Cont.*

| Variable | N (%) |
|---|---|
| Highest degree or level of education | |
| Some high school | 1 (6.7) |
| Completed high school | 1 (6.7) |
| TAFE | 4 (26.7) |
| Undergraduate degree | 4 (26.7) |
| Postgraduate degree | 5 (33.3) |
| Current employment status | |
| Employed full time | 3 (20.0) |
| Employed part time | 1 (6.7) |
| Unemployed | 1 (6.7) |
| Retired | 9 (60.0) |
| Carer's leave | 1 (6.7) |
| Participants' current exercise type | |
| Exercise professional lead program | 5 (35.7) |
| Structured regular exercise without exercise health professional | 2 (14.3) |
| Unstructured incidental exercise | 5 (35.7) |
| Not completing regular exercise | 2 (14.3) |

*3.2. Summary of Results*

Ten participants reviewed synthesized member checking summaries: seven participants made no changes, three made minor additions, and five did not respond (Figure 1). The mean (SD) interview duration was 63.8 (20.5) minutes, and the mean focus group duration was 68 (11.1) minutes.

Outcomes of importance reported by people with lung cancer and a carer of someone with lung cancer spanned all five domains of the ICF framework. Two additional categories related to outcomes of importance were related to multiple ICF domains that were identified by participants.

*3.3. Outcomes That Were Classified in the ICF Framework*

There were 63 outcomes within the ICF framework of exercise programs reported by participants to be important to them, and these were related to activity and participation (*n* = 24, 46%), body function (*n* = 19, 37%), environmental factors (*n* = 5, 10%), body structure (*n* = 3, 6%), and personal factors (*n* = 1, 2%) (Figure 2).

Within the activity and participation domain, reported outcomes spanned all but one of the nine chapters (89%). Over one-third (*n* = 9/24, 38%) of reported ICF outcomes identified were related to the mobility chapter of the activity and participation domain. Comparatively, only five of the eight (63%) chapters in the body functions domain and three of the five (60%) chapters of the environmental factors domain were identified as important to participants. Within the body functions domain, almost half of the outcomes reported (*n* = 8/19, 42%) were linked to the chapter on functions of the cardiovascular, hematological, immunological, and respiratory systems. The lowest proportion of chapters were perceived to be important to participants in the body structure domain, with three of the eight chapters (38%) linked to ICF outcomes during deductive analysis (see Figure S1 for full details of outcomes of importance and corresponding codes). Personal factors were coded inductively using conventional content analysis and included in Figure 2 and the manuscript text; however, they were not included in Figure S1, as this domain of the ICF does not have defined chapters and codes in the hierarchy of classifications [11].

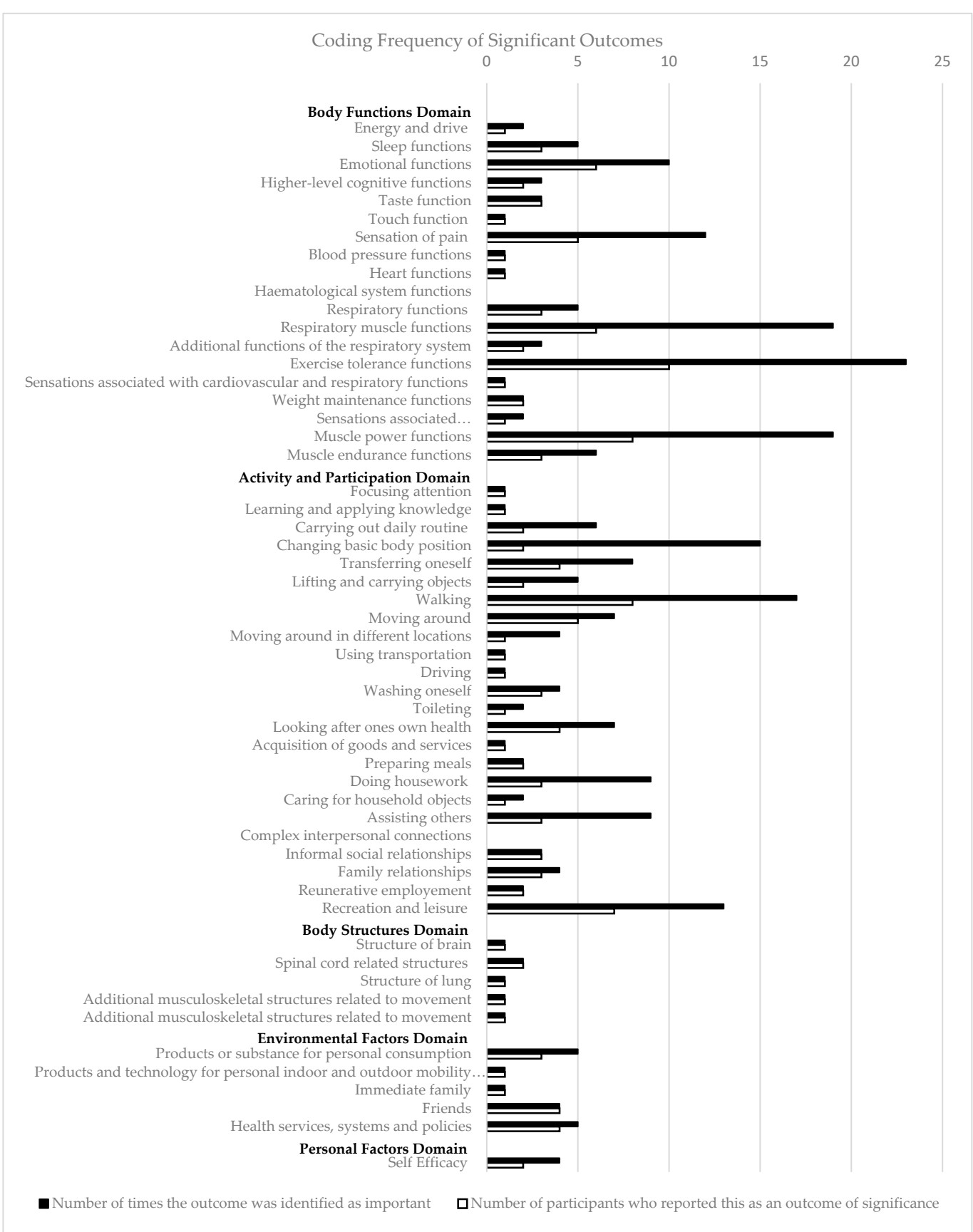

**Figure 2.** Coding frequency of significant outcomes.

A minimal difference was observed (under 3%) between the number of ICF level 2 codes counted and the number of counted words attributed to the personal factors, body structures, and environmental factors domains (Figure 3). A difference of 11% was seen between the volume of words (46%) and the number of ICF codes (35%) attributed to each domain.

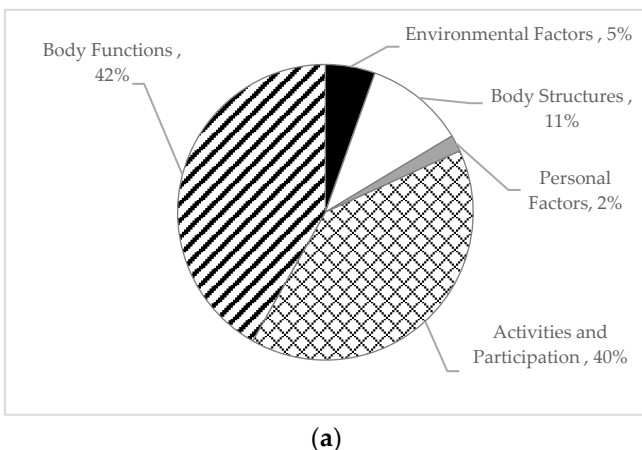
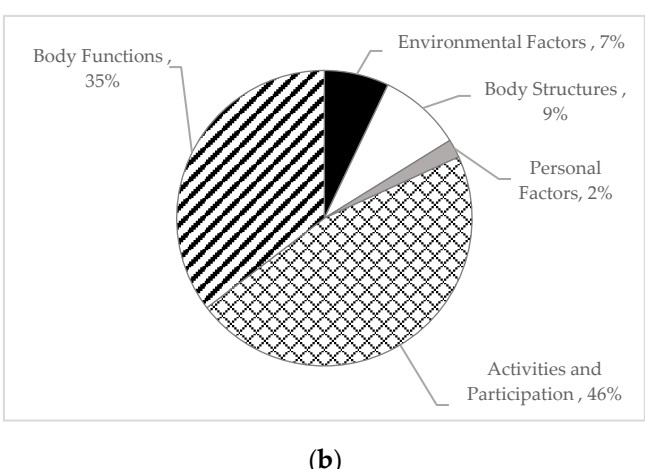

(**a**) (**b**)

**Figure 3.** Percentage of ICF codes and participant quotes attributed to each ICF domain: (**a**) number of words attributed to each domain; (**b**) number of ICF codes attributed to each domain.

### 3.3.1. Domain 1: Activity and Participation

Participants highlighted the ability to walk and maintain balance as outcomes of high priority.

> "The walking is the big one actually. If you can walk you put a bit of condition on your leg, some balance, you keep your balance... To me walking with confidence." (Participant 13)

Being able to mobilize outside of the home environment and use transportation, i.e., public transport, was important to the participants.

> "I'm a single person... it's the ability to go out into the world is the most important thing." (Participant 11)

> "I was going in and I was getting on the tram and going up to the gym . . . These small measures became important to me." (Participant 12)

Measuring a person's current ability to perform tasks essential for self-care or caring for children or family members was consistently reported as a high-importance outcome for participants with caring responsibilities.

> "I wanted them to understand that I have to lift. I have to lift kids from a bath. Kids who can't stand up and walk yet . . . I wanted them to try and understand the movements I have to do." (Participant 7)

Participants emphasized the importance of measuring a person's ability to look after their own physical and mental health.

> "I think you should ask them what they do to look after themselves . . . what is it they do about looking after their body? and also not to be separated from their mind. Because you can't separate the two." (Participant 11)

### 3.3.2. Domain 2: Body Function

Breathlessness, specifically during exertion, was highlighted by participants as an exercise outcome of importance.

> "The shortness of breath... like the minute I would start to walk uphill it would be impacted straightaway." (Participant 3)

Cognition was highlighted as an outcome that was under-measured in the experience of participants.

> "The cognitive impact is number two and I think it's not measured enough." (Participant 7)

Participants linked body function outcomes to activity and participation domains, for example, the importance of cognition for enhancing family relationships.

> "Being able to sit and concentrate and do some drawing and art with my daughter and feeling comfortable with that." (Participant 14)

The importance of muscle power in the upper and lower body was regarded by participants as an essential component for completing daily tasks.

> "Just having the ability, having enough strength to be able to just do stuff without feeling like oh God I've got to go up the flight of stairs, here we go again." (Participant 5)

It was important to participants to be asked about the onset and specific characteristics of symptoms they may be experiencing.

> "Doing an activity and then saying 'can you rate your muscle pain', like for me with this drug where you do get muscle pain, what's your muscle pain after you do an activity?" (Participant 14)

### 3.3.3. Domain 3: Body Structure

Participants emphasized the importance of understanding the connection between disease spread and body function changes.

> "I've got brain mets as well, I think this is going to be, there needs to be a little bit more involvement of the cognitive side effects." (Participant 14)

Measuring treatment-related structural changes was important to participants for ensuring exercise interventions were tailored and individualized.

> "It's really important to know what my bone strength, and my bone density is like because we don't know the impact of these treatments on those sort of things then I know what exercises I need to tailor to improve." (Participant 14)

### 3.3.4. Domain 4: Environmental Factors

Participants highlighted the importance of measuring support and relationships, as it is an outcome that is both personally important and may impact their ability to perform an exercise program.

> "Whether the person has got support and is you know engaged in the community or in their life in general . . . I think that would impact how you might participate in your exercise program." (Participant 4)

A common outcome of significance was the side effects and symptoms of the medications or anti-cancer treatments a person was taking.

> "My struggle is the side effects of the medication. . . that that's my barrier with exercise." (Participant 18)

### 3.3.5. Domain 5: Personal Factors

Participants reported self-efficacy as an outcome of significance for people with lung cancer participating in an exercise program.

> "A lot of it's mentally being confident that I can build myself back up because you get so many knocks when you are going through your cancer journey, and you've got so little control over that aspect of your life." (Participant 15)

*3.4. Outcomes That Span Multiple ICF Framework Domains*

3.4.1. Barriers to Exercise

It was important to participants that exercise professionals identified barriers that may limit their capacity to perform exercise. Barriers identified by participants included identifying medications, medication associated symptoms, and physical ability.

> "It's those symptoms, it's knowing OK with the drugs that she's on, they cause muscle fatigue and pain and dysfunction so at what point is that happening with certain exercises? So that we could try and overcome that." (Participant 14)

3.4.2. Impact of Age, Frailty, and Comorbidities

Participants spoke about how clinician understanding of the impact of age, frailty, and comorbidities was vital as it impacted their ability to exercise.

> "It's a combination of the double whammy with old age. The fact that I now have cardiotoxicity, and I have got extremely low blood pressure." (Participant 7)

> "If you've got somebody who's more frail or having other treatment . . . I think it would be important to discuss it." (Participant 4)

*3.5. Codes Outside of the ICF Framework Domains Regarding the Selection, Use, and Completion of Outcome Measures*

3.5.1. Individualized and Tailored Outcomes

Participants emphasized the importance of achieving tailored and individualized outcomes. This was important when selecting contextually relevant outcomes for different patients.

> "Everybody is going to have a different answer. Depending on what sort of treatment they have. . . . It might have been a small lung cancer, it could have been surgery, it could have been anything because they are all so different." (Participant 10)

Tailored and individualized outcomes were important when measuring outcomes for the same patient at different stages of their cancer trajectory.

> "And it changed as I went along. Like I have been diagnosed seven and half years ago . . . I might have answered those questions for the whole thing, but like my experiences changed throughout. It's not. This has not been the way it's been from the start to now." (Participant 14)

3.5.2. Importance of Explaining Outcome Measures to Patients

Participants with experience completing the outcome measures in a clinical setting reported frustration when they did not receive feedback after completing an outcome measure.

> "Going through the health system, you come across many questionnaires and patient outcome measures . . . but too often, once you fill them in diligently, there's no feedback." (Participant 5)

Measuring specific exercise metrics when examining exercise ability and completion in order to relay this information back to the participant was reported to be important.

> "I was really disappointed . . . I used to get on my bike . . . and I would think each time he (physiotherapist) was writing down how far I'd pedaled . . . until I saw his chart and he was just ticking whether I'd done it or not." (Participant 16)

Participants also highlighted the importance of exercise professionals explaining changes in outcomes.

> "You don't really see objectively yourself so much at home how much you have improved than if you have someone who is saying do you realize that three months ago you couldn't even do it three crouches. So this process is really important." (Participant 11)

## 4. Discussion

This qualitative study identified 63 ICF framework outcome measures in the context of exercise that are important to people living with lung cancer and carers. The most frequently identified domain was "Activity and Participation", with over one-third of reported ICF outcome codes attributable to this domain. Within this domain, chapters relating to mobility, activities of daily living, and social connection were the most frequently referenced by participants. The importance of this domain is emphasized in Patient-Reported Outcome Measures (PROM), with 97% of PROM items used in the oncology setting attributable to the activity and participation domain [28]. This is consistent with previous qualitative research exploring patient perspectives on supportive care needs and health care experiences, where people with lung cancer highlight the importance of living an everyday life and completing activities of daily living [29].

The domains of mobility, self-care, and domestic life (Chapters 4, 5, and 6) are the most frequently measured in oncology PROMs and generic health-related quality of life (HRQoL) outcome measures [28,30]. Furthermore, the activity and participation domain of the ICF framework has been found to be the most frequently measured in both single-domain and multi-domain frailty measures [31].

Qualitative research reports that symptom distress is often relative to its impact on activities of normal life rather than symptom intensity or duration [32]. Our findings show that some participants reported important outcomes within the body functions domain that were linked to the activity and participation domain. One example was having the upper body strength to be able to care for family members. The centrality of the impact on activity and participation outcomes when measuring meaningful outcomes was further reflected in outcomes linked to the environmental domain, where products and technology for personal indoor and outdoor mobility and transportation were linked to walking and socializing.

The most frequently reported outcome was exercise tolerance functions and respiratory muscle functions (Figure 2). We highlighted that outcome measures related to physical functioning that are commonly reported in lung cancer exercise research [33], i.e., aerobic capacity and muscle strength, are also important to people with lung cancer and their carers. This is further supported by research exploring patient experiences when participating in exercise interventions, where positive experiences are described within the context of improved physical functioning and decreased symptom burden [34,35].

The least frequently referenced primary domain was body structures, with the majority of quotes attributable to this domain made by a single participant. Outcomes linked to the body structures domain appear to be of greater importance to clinical experts than healthcare consumers. This is in agreement with research describing outcomes of significance for falls risk assessment in older adults, where body structure outcomes made up 6% of significant outcomes for consumers [13] compared to 18% for healthcare practitioners [36].

Outcomes of importance that are not regularly used in current lung cancer and exercise research [33,37] or recommended in pulmonary rehabilitation recommendations [38] are cognition and specific pain features, including pain frequency, impact on activity, and medication use. These outcomes have been described as unmet supportive care needs by people with lung cancer post-chemotherapy and post-thoracic surgery [39]. These specific participant concerns are not measured in oncology PROMs [28] or lung cancer patient-reported quality of life questionnaires [40,41].

Outside of the ICF framework, participants emphasized the importance of clinical communication around outcome measure collection to improve their understanding of what and why outcomes were collected. Participants spoke about how understanding changes in outcomes regarding exercise ability motivated them to continue to exercise. The clinical communication needs of patients with cancer have been shown to vary across disease stages, and depending on individual patients' preferences [42], clinicians should consider individualizing and modifying feedback on outcome measures to patients.

*Strengths and Limitations*

This is the first qualitative study describing outcomes of exercise programs that are important from the perspectives of people with lung cancer and their carers. Using virtual modes for participation enabled greater representation of people with lung cancer from rural and regional areas, as well as multiple Australian states.

The use of conventional content analysis enabled researchers to identify important outcomes that did not clearly fall within the ICF framework. For example, measuring frailty was important to participants; however, frailty cannot be coded to a single ICF domain or chapter. The use of directed content analysis following inductive analysis enabled the details of a coded outcome to be mapped to all relevant chapters of the ICF to 'unpack' the potentially measurable components of this outcome. For example, 'transfers and mobility' was an inductive code generated by researchers (A.B. and L.E.), and when deductively mapped to the ICF framework, this was linked to seven separate outcome codes (changing basic body position from sitting, changing basic body position from standing, transferring oneself, walking short distances, walking long distances, moving around within the home, and moving around within buildings other than the home). In this example, seven separate outcome codes were identified for the consideration of use in clinical practice to ensure that all components of the outcomes that were important to patients were identified.

Several study limitations should be acknowledged. As reported in previous exercise research studies in lung cancer, people who have an interest in or have previously participated in an exercise program were more likely to demonstrate interest in this study. Each participant reported varied experience with exercise programs, consistent with the real-world experiences of the general population.

The results of this paper largely represent the views of English-speaking people with lung cancer who were performing supervised or unsupervised exercise programs. Therefore, these results may not be representative of outcomes of significance for carers of people with lung cancer, people from a culturally and linguistically diverse background, or those with lung cancer who have not previously engaged in exercise.

Only one carer participated in a focus group, so while the aim of this study was to explore outcomes of significance for people with lung cancer and carers of people with lung cancer, with only one carer being included, the results of this study do not broadly represent the views of carers of people with lung cancer.

## 5. Conclusions

This qualitative study provides a consumer perspective on exercise outcome measure collection in the setting of exercise assessment, education, and intervention, highlighting both the outcomes that are important to people with lung cancer and elaborating on how the collection and communication of these outcomes impacts the consumer experience.

Outcomes used to evaluate lung cancer exercise programs should be tailored and individualized to each patient. When selecting outcome measures, clinicians and researchers should consider selecting outcomes from all relevant domains of the ICF, with a focus on the impact to activity and participation, in addition to non-cancer factors such as ageing, frailty, and co-morbidities. Clinicians should explain and discuss outcome measure findings and changes with patients. Future research should include developing a patient-centered core outcome set to integrate a standardized set of patient-relevant outcomes into clinical practice and enable further research in the lung cancer and exercise setting.

**Supplementary Materials:** The following supporting information can be downloaded via this link: https://www.mdpi.com/article/10.3390/curroncol31020054/s1, Section S1: Question Guide; Figure S1: ICF framework domains, chapters, and outcomes.

**Author Contributions:** Conceptualization, A.B., L.E. and L.D.; methodology, A.B., L.E., L.D. and C.E.M.; formal analysis, A.B. and L.E.; data curation, A.B.; writing—original draft preparation, A.B.; writing—review and editing, A.B., L.E., L.D. and C.E.M.; visualization, A.B., L.E., L.D. and C.E.M.; supervision, L.E. and L.D.; funding acquisition, L.E. All authors have read and agreed to the published version of the manuscript.

**Funding:** This research was funded by the University of Melbourne, grant number 2021ECR097.

**Institutional Review Board Statement:** The study was approved by the University of Melbourne Human Research Ethics Committee (Project Number: 2021-22492-20870-3).

**Informed Consent Statement:** Informed consent was obtained from all subjects involved in this study.

**Data Availability Statement:** The data presented in this study are available on request from the corresponding author. The data are not publicly available due to privacy and confidentiality considerations.

**Conflicts of Interest:** The authors declare no conflicts of interest.

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
