# Peer review of "“The Ability to Go Out into the World Is the Most Important Thing”—A Qualitative Study of Important Exercise Outcomes for People with Lung Cancer"

_curroncol, doi:10.3390/curroncol31020054_

Round 1
Reviewer 1 Report
Comments and Suggestions for Authors
Overall this paper makes an important contribution to the literature relating to measurement of outcomes following exercise based rehabilitation for people with lung cancer. It reports a robustly conducted study within a well described qualitative design appropriate to the research aims.
I would suggest a couple of amendments to strengthen the reporting of your study and findings.
Methods:
2.3 Study design Lines 116-132
You state that the ICF was used to inform development of your question guide, but what epistemological /theoretical position underpins the whole study design? It reads like it was informed by a pragmatist approach as described by Cornish and Gillespie in 2009.
Can you explain why content analysis was chosen over more inductive methodologies? How is content analysis compatible with the inductive phase of your analysis?
2.4. Participants, recruitment, and sample. Lines 133-141 AND Strengths and Limitations page 12 from 398:
Did you consider purposive sampling given the heterogenous clinical and socio-demographic characteristics of people diagnosed with lung cancer who may benefit from participating in a variety of modes of exercise? If not, might this be discussed as a limitation? You report that most participants were already participating or had participated in exercise so is there a potential for selection bias that might have been overcome with a purposive sampling design? Also, 50% of the participants had received surgical treatment- how confident can we be that these outcomes would be important to the larger proportion of people with lung cancer who are not treated surgically?
Discussion:
There is some repetition of the findings in the discussion section.
While I appreciate that this study is focused on what outcomes are important to patients relating to exercise rehabilitation, the paper would be strengthened if more word count in the discussion section was given to relating the findings to the wider lung cancer qualitative literature relating to peoples needs and experiences of cancer treatment, rehab and exercise. e.g., Salander, P., & Lilliehorn, S. (2016) and others.
Author Response
Thank you very much for taking the time to review this manuscript. Our responses and revisions, which we feel have strengthened our submission, are provided in the tables below. Revisions are highlighted in red and text that has been moved in the text in green.

Reviewer 2 Report
Comments and Suggestions for Authors
I especially liked how the authors used a qualitative study to investigate the outcomes of exercise that are significant to people with lung cancer and their carers. While there is a large body of literature on the effectiveness of healthcare interventions in managing the symptoms of cancer patients, less is known about patient perspectives and experiences with the impact of healthcare interventions. In this way, the authors were able to identify the real gap in the literature on this topic for addressing patients’ needs. Overall, the study is very well presented, and how the authors decided to analyse and present data is very interesting. I only have a few small concerns to address. Please find attached my specific comments.

Author Response
Thank you for taking the time reviewing our manuscript and the questions and comments provided. Our responses and revisions, which we feel have strengthened our submission, are provided in the tables below and the revised manuscript. Revisions are highlighted in red and text that has been moved within the text in green.

Round 2
Reviewer 1 Report
Comments and Suggestions for Authors
I enjoyed reading the revised manuscript and think it makes an important contribution to the science of rehabilitation in cancer.
Reviewer 2 Report
Comments and Suggestions for Authors
The authors have appropriately addressed all my concerns.